# Performance of a Smart Device over 12-Months for Home Monitoring of Patients with Intermediate Age-Related Macular Degeneration

**DOI:** 10.3390/jcm12072530

**Published:** 2023-03-27

**Authors:** Selwyn Prea, Robyn Guymer, George Kong, Algis Vingrys

**Affiliations:** 1Department of Optometry and Vision Sciences, The University of Melbourne, Melbourne 3010, Australia; 2Royal Victorian Eye and Ear Hospital, East Melbourne 3002, Australia; 3Centre for Eye Research Australia, Royal Victorian Eye and Ear Hospital, East Melbourne 3002, Australia; 4Ophthalmology, Department of Surgery, The University of Melbourne, Melbourne 3010, Australia

**Keywords:** telemedicine, home-monitoring, visual fields, microperimetry, age-related macular degeneration, tablet device, iPad

## Abstract

Background: To determine the 12-month compliance with and retention of home monitoring (HM) with Melbourne Rapid Fields (MRFh) for patients with intermediate age-related macular degeneration (iAMD) and compare visual acuity (VA) and retinal sensitivity (RS) results to clinical measures. Methods: Participants were recruited to a 12-month HM study with weekly testing of vision with MRFh. Inclusion criteria were a diagnosis of iAMD, understand English instructions, VA ≥ 20/40, and access to an iPad. Supervised in-clinic testing of high contrast VA (HVA, ETDRS), low-luminance VA (LLVA, ETDRS with ND2 filter), and RS (Macular Integrity Assessment, MAIA, and MRF in-clinic, MRFc) was conducted every 6-months. Results: A total of 54 participants (67 ± 6.8 years) were enrolled. Compliance to weekly HM was 61% and study retention at 12-months was 50% of those with uptake (*n* = 46). No difference was observed between MRFc and MRFh across all RS and VA outcomes (*p* > 0.05). MRFh RS was higher than MAIA (29.1 vs. 27.1 dB, *p* < 0.001). MRFh HVA was not different from ETDRS (*p* = 0.08), but LLVA was 9 letters better (81.5 vs. 72.4 letters, *p* < 0.001). Conclusions: Over 12-months, MRFh yields a moderate level of compliance with (61%) and retention (50%) of weekly testing. Further studies are required to assess the ability of MRFh to detect early progression to nAMD.

## 1. Introduction

Intravitreal injections of anti-vascular endothelial growth factor (anti-VEGF) have become the mainstay treatment to halt vision loss in patients with neovascular age-related macular degeneration (nAMD). One of the main clinical challenges rests in the timely detection of conversion from intermediate AMD (iAMD) to nAMD so that treatment can be started with minimal delay to optimize the chance of achieving good visual results [1]. Between scheduled hospital visits, the main strategy to detect new onset of nAMD is home monitoring with an Amsler grid. This method dates back to 1945 [2] and shows poor sensitivity to nAMD detection [3]. There is an enormous unmet need to provide better home monitoring (HM) solutions, and considerable research is being undertaken to fill it.

A cost-effective approach may be to use smart devices, such as phones and tablets, which are ubiquitous and familiar to patients; this makes them an appealing platform for home monitoring. However, to date, pixel densities do not allow the measurement of hyperacuity thresholds at the usual reading distance of 40 cm (16 inches), limiting their application for threshold measurements [4]. Retinal sensitivity (RS) at the macula may be another useful parameter to consider when aiming to detect the progression from iAMD to nAMD. Testing RS does not require high levels of pixel resolution and can be achieved on smart technology. It may be possible to detect macular sensitivity losses before there are changes in visual acuity [5], and previous reports show correlations between macular pathology seen with optical coherence tomography (OCT) and RS [6]. Furthermore, RS measured with a tablet device at home, over the short-term (from 1.5 to 2 months), is comparable to that measured by the Macular Integrity Assessment (MAIA, CenterVue, Padova, Italy) during supervised in-clinic testing [7,8,9].

Another issue with any device or technique designed to be used in the home for long-term monitoring of visual function is compliance. It is also important for the home monitoring technique to record results in the home that are comparable to those recorded in clinic to ensure validity. Melbourne Rapid Fields (MRF, Glance Optical, Melbourne, Australia) is an application that facilitates VA and macula RS testing on an iPad (Apple, Cupertino, CA, USA). High contrast visual acuity (HVA) and low luminance visual acuity (LLVA), in addition to macula RS, can be tested at home using application-generated voice prompts. In a cohort of stable AMD [9] or glaucoma patients [10], good compliance and test reliability was found in the short-term (weekly testing for 6-weeks). The present study aims to investigate iAMD patient’s longer-term (12-month) compliance with and retention of HM using the MRF (MRFh). A secondary aim is to compare MRFh outcomes to those taken on the same device in the clinic (MRFc) as well as to standard clinical testing of VA (EDTRS chart) and RS (MAIA).

## 2. Materials and Methods

Ethics approval was obtained from the Royal Victorian Eye and Ear Hospital ethics committee (AMD: HREC: 95/238H/15). All experiments were conducted in accordance with the tenets of the Declaration of Helsinki, and all participants were required to give informed consent prior to taking part in this study. All clinical and management decisions were determined using routine clinical methods, and the MRF was run in parallel but did not contribute to clinical decision making. MRF clinical data was analysed post-hoc and only after the study was completed, whilst early uptake and compliance data has already been analysed and reported elsewhere [9].

### 2.1. Participants

Participants with intermediate AMD (iAMD) were recruited from the Macular Research Unit of the Centre for Eye Research Australia (CERA, East Melbourne, Australia). These participants provided data for the short-term (6-week) compliance study [9], and the same participants continued testing out to 12 months. Inclusion criteria were bilateral large drusen (≥125 µm) (Beckman classification of iAMD), visual acuity better than or equal to 20/40, access to a compatible smart device (Apple iPad Gen 4 or newer, Apple iPad Air Gen1 or newer, Apple iPad Mini, Apple, Cupertino, CA, USA, not supplied by our research group) with active internet connection, and the ability to understand English commands as provided by the iPad audio prompts. Participants were excluded if they had evidence of late-stage AMD (geographic atrophy or nAMD) at enrolment or if they had undergone recent intraocular surgery (past 6-months).

### 2.2. Melbourne Rapid Fields Vision Testing Application

The Melbourne Rapid Fields-macular iPad application used for this HM study has been detailed elsewhere [9], and in this trial it was modified to work on an iPad Mini as well as the standard 9.7-inch version of the Apple device. The iPad was chosen due to the superior Apple Retina display that made testing of RS and VA possible. Visual acuity is thresholded using 2 different Landolt C optotypes with interaction bars; high contrast visual acuity (HVA, L_b_: 135 cd/m^2^, contrast: 99%, Figure 1(ai)), and low-luminance visual acuity (LLVA, L_b_: 5 cd/m^2^, contrast: 84%, Figure 1(aii)). Participants respond via a four alternate-forced-choice protocol (Figure 1b). Visual acuity requires a matching task where participants identify the orientation of a Landolt C target and tap their choice from four possible alternatives that are shown underneath the target (Figure 1b). Letter size decreases by one step (0.1 logMAR) once correct identification (two out of three) takes place, and the final acuity is returned for the last line where two out of three are seen correctly. LogMAR values are converted to an ETDRS letter score using the formula of $\mathrm{ETDRS}=\frac{1.7-\mathrm{logMAR}}{0.02}$ for comparison against the ETDRS chart [11]. All MRF acuity testing is performed at 33 cm with the participant wearing their normal reading glasses. The clinical procedure for LLVA is to use a 2 ND filter for testing, whereas the MRF background changes over 1.45 ND to 5 cd/m^2^, being somewhat brighter than the clinical test (1.3 cd/m^2^).

Visual field testing with the MRF measures RS in the central 9.5° with a 33-point, size-scaled, radial grid similar to the MAIA (Figure 1c). Threshold is achieved on a 5 cd/m^2^ background with a 3-step Bayes approach and a neighbourhood logic that inserts extra test points to the test grid when values deviate from age-expected values. These added test points do not contribute to the global indices but are used to define the spatial extent of any lesion. The user is guided through the test via audio prompts that provide instructions on taking the VA and VF tests (Figure 1d) and saving the test to the online portal. HVA is tested first, followed by LLVA, and the VF test is completed last. The participant is asked to wear their reading glasses, and the test is conducted at a viewing distance of 33 cm. Testing with MRF was performed in clinic under the supervision of the study coordinator (MRFc) and at home with application-generated audio prompts (MRFh), and results were compared.

The MRF application tracks VF test reliability by monitoring false positive (FP) and false negative (FN) responses. A false positive error is logged when a response is made with no stimulus present. A FN error occurs when a response is not made to a previously seen threshold level. Whilst fixation tracking is not a built-in feature of the application, the voice prompt periodically reminds the user to fixate on the centre of the red cross fixation target. The criteria for a reliable test were taken as FPs and FNs ≤ 25%.

### 2.3. Testing Procedures

Figure 2 outlines the testing protocol for this 12-month study. Participants with iAMD who were involved in other natural history studies at The Centre for Eye Research Australia (CERA) and owned an iPad (mini or larger) [9] were recruited to this study. AMD classification was made by a reading centre based on colour fundus photography. After consent was given at the recruitment visit, the study coordinator set up the application on the participant’s device. Training was provided on how to use the application in the form of oral and written instructions, including how to save and submit test results online. Each clinical visit included Early Treatment Diabetic Retinopathy Study (ETDRS) VA (HVA and LLVA with ND2 filter) measured at 20 ft (6 m), OCT, fundus photography, microperimetry (MAIA), and MRFc testing (Figure 2). Between the scheduled 6-monthly clinical reviews, participants were asked to perform weekly HM (MRFh) in their home, which included VA testing (HVA and LLVA, at near) and RS. At the 12-month completion of the study, a survey of participant perceptions of the home monitoring was undertaken (Figure 2). Ad hoc clinic visits were undertaken if the participants developed new symptoms or as determined by clinical need.

### 2.4. Data Analysis

Participants with iAMD undertook MRF testing on both eyes, with the right eye tested first. The study investigators analysed data from the best eye, which was classified as the eye with the highest mean RS on MAIA microperimetry at baseline. For instances where the mean RS was equal between the eyes, the right eye was selected for analysis. The average number of reliable tests was calculated for each participant, and test methods were compared with a *t*-test. All data are reported as mean (±standard deviation). The mean absolute error (MAE) was calculated over the first 26 weeks of MRFh for the median value of RS (median less affected by outliers).

The MRF application tracks VF test reliability by monitoring false positive (FP) and false negative (FN) responses. Whilst fixation tracking is not a built-in feature of the application, the voice prompt periodically reminds the user to fixate on the centre of the red cross fixation target. The criteria for a reliable test was for FP and FN errors to be ≤25%. The criteria for a reliable test on the MAIA was FPs ≤ 25%.

Participants were advised to set their own reminders to complete the test on a day and time that suited them best and to perform testing in the same quiet room with the same lighting conditions. Weekly test reminders were not provided, however, two weeks after a missed test submission, an automated email reminder was sent to the participant. Compliance was considered in two ways: the first identified a MRFh examination returned within 7 + 1 days of the last test (weekly schedule), and the second identified those returned within a week after the reminder email (14 + 7 days).

At study completion, a survey of perceptions of home monitoring that comprised nine questions was given to the uptake group. The questions were scored using a 5-point Likert scale (strongly disagree, disagree, neutral, agree, strongly agree) and significance was achieved when the group average was significantly removed from the neutral state of 2.5. The survey questions were based on experience of the researchers and were not from a validated questionnaire. The questions were as follows:

Q1: Overall, the MRF application was easy to use.

Q2: I found the voice commands helpful.

Q3: I felt confident opening the application and logging in.

Q4: I felt confident performing the visual field test.

Q5: I felt confident saving and submitting my test result.

Q6: I felt confident knowing which eye to test.

Q7: I am likely to comply to weekly testing with the MRFh for 1 year.

Q8: I am likely to comply to monthly testing with MRFh for 1 year.

Q9: I felt the iPad was more comfortable compared to the perimeter in clinic (MAIA).

## 3. Results

### 3.1. Compliance and Retention to 12-Months of MRFh

A total of 54 participants were recruited for the study, with a mean age of 67 years (51–82) and 82% being female (Table 1). Eight participants did not return any MRFh test results (Dropout, Figure 3), and as such were excluded from further analysis. Subsequent interviews identified the following reasons for dropout: lack of motivation/too much effort (75%, *n* = 6), MRFh difficult to use (13%, *n* = 1), and competing life demands (13%, *n* = 1). The remaining 46 participants successfully submitted ≥1 home examination (Figure 3).

Of the expected 2392 (46 × 52) MRFh tests that should have taken place over 52 weeks, 1338 (56%) were returned from *n* = 46 participants with uptake. Of the 1338 tests received, 61% were submitted within a week (7 + 1 days, Table 2), and 82% of tests were submitted within 2 weeks (14 + 1 days, Table 2) of the previous test submission, which did not prompt a reminder email. After two weeks of inactivity (>15 days), a reminder email was sent, and compliance improved to 92% of tests being submitted within three weeks of their intended date (Table 2). After a period of four weeks (>29 days), 96% of tests were received. Taking all tests received into consideration, a test result was received on average once every 9.9 days.

In the uptake group of *n* = 46 participants, 83% (*n* = 38, Figure 4a) were active at 26 weeks and 50% were retained at the end of the study (*n* = 23, Figure 4a). The average compliance to the request for weekly MRFh was 61%, and this varied across the year as participants dropped out (dotted line, Figure 4b, Table 2).

Test reliability is summarized in Table 3. MRFh was not significantly different from MRFc (*p* = 0.06, Table 3), and MRFc was not different from the MAIA (*p* = 0.32, Table 3). Test reliability from MRFh was significantly poorer than the MAIA (*p* < 0.001, Table 3). The average test time was 5.3 ± 0.3 min for the MAIA and 1.7 ± 0.4 min for MRFh, which is reported elsewhere [9].

In 11% (*n* = 5/46), a low RS (>95% departure from their median RS value) was observed on the following test submission, and in 7% (*n* = 3/46), a low RS was seen twice in a row, generating an alert to the study coordinator. At the end of the study, there was no clinical evidence of progression in any of our participants, indicating that the rate of false positive alerts generated by the device was 7% for the uptake group.

### 3.2. Comparison of Standard Clinical Assessment of RS and VA to MRFc and MRFh

The MAE for MRFh RS was compared to that of the MAIA over the first 26 weeks of MRFh (Figure 5). The MRFh MAE (triangles, Figure 5) was not significantly different from the MAIA at baseline and by week 6 (circles, Figure 5, *p* = 0.16).

Table 4 compares in-clinic assays of VA and VF (ETDRS and MAIA) to the equivalent tablet test performed with MRFc or MRFh. A significantly better RS was returned from MRFc (*p* < 0.001, Table 4) and from MRFh (*p* < 0.001, Table 4) compared to the in-clinic MAIA. There was no difference between the HVA tests measured at home or in-clinic (*p* > 0.05, Table 4). LLVA was approximately 9 letters better with MRF than when using the ETDRS with a ND2 filter (*p* < 0.001, Table 4).

When comparing supervised MRFc to MRFh, there was no significant differences observed across all results of RS and VA (Table 4, *p* > 0.05).

### 3.3. Survey of 12-Months of Home Monitoring

Survey results were received from *n* = 40 participants with iAMD with HM uptake. Overall, the responses were positive and in favour of the new technology, with the MRF being considered easy to use (4.6, *p* < 0.01, Q1, Figure 6) and the audio commands assisting with testing (3.4, *p* < 0.01, Q2, Figure 6). Furthermore, the majority of participants stated they were likely to continue to comply with weekly testing in the longer-term (3.9, *p* < 0.01, Q7, Figure 6), and that they felt visual field testing with the tablet was more comfortable than the MAIA microperimeter performed in the clinic (4.7, *p* < 0.01, Q9, Figure 6).

## 4. Discussion

Long term compliance with home monitoring is challenging but would likely improve early detection of the onset of neovascular AMD if those at high risk of progression could regularly test their vision. We present the findings of a 12-month home monitoring study in participants with iAMD, including compliance and comparison to standard clinical testing. We find compliance to the request for weekly MRFh to be 61% in the absence of test reminders. This is less than our finding of 72% for glaucoma patients who also monitored their visual fields weekly with MRF over the same timeframe, but who received personalized text message reminders from the study coordinator on the day a test was due [10]. In the current study, we report that sending out email reminders after 2 weeks of inactivity improved compliance by 10%. Our findings are in line with other long-term home monitoring trials that used the purpose-built preferential hyperacuity perimetry (PHP) device and found adequate compliance in the absence of test reminders [2,12,13].

Hyperacuity is the preferred testing modality for monitoring patients at home. Hyperacuity is the ability to resolve the difference in relative spatial location between two or more stimuli and has been put forward as a suitable parameter for monitoring macula disease due to the minimal effect that ageing has on results [14,15]. A clinical trial where iAMD patients undertook daily HM with a PHP device (Notal ForeseeHome, Tel Aviv, Israel) found that the VA of participants who converted to nAMD was approximately one line better at presentation in the device group [12]. Over a mean follow-up period of 1.4 years, test compliance was 63%, even though daily rather than weekly testing was requested [12]. In another real-world HM review of company records over a 9-year period with the same device, daily compliance was reported at 80%, with 69% of conversions to nAMD detected by system alerts [13]. Similarly, home OCT studies have shown that nAMD patients undergoing anti-VEGF treatment are capable of returning high quality OCT images from home with a strong correlation to human and automated grading of retinal fluid [16]. However, HM with devices such as the PHP and home-OCT require purpose-built hardware that has a significant cost and requires considerable support to ensure it is delivered efficiently, making its utility not readily accessible globally.

We did find that only 46 of the initial 54 participants recruited to our study (85%) performed 1 or more tests in the year. The major reason for participant dropout was lack of motivation. Furthermore, our active group reduced to *n* = 23 (50%) by the end of the study, which suggests that long term monitoring of people with iAMD may well be very difficult, especially when monitoring is likely to be for significantly longer periods than 12 months. Others have suggested that compliance might be improved with gamification, where fun and engaging targets and feedback are blended with the clinical test to maintain compliance [17,18]. While there is some evidence to show that gamification can increase compliance rates [18], there is also evidence that gamifications require longer test times that can have deleterious effects on thresholds [19,20], and there is scant data on the benefits of gamification in geriatric populations [21]. Although the authors of the present study did not observe a relationship between older age and poor compliance/test reliability, a larger clinical trial is warranted to fully investigate associations with poor compliance.

Despite our average compliance finding of 61%, we believe weekly testing is the optimal frequency for high-risk AMD patients. We find that people with AMD are familiar with undertaking Amsler grid testing on a weekly basis to detect early changes. The findings of a HM simulation by Anderson et al. [22] demonstrate substantial benefits in detecting change with weekly or fortnightly testing over six months of in-clinic reviews, even in the presence of modest compliance (63%). Another way to consider this issue is from the patient’s perspective. Our compliance rate indicates that, even with the level that we found (61%), we can expect to receive a result once every 2 weeks, which is still adequate for detecting change in the majority of people (82% of test results received every 2 weeks, Table 2). We do not think monthly testing provides an adequate frequency for monitoring acute vision change. On a monthly schedule, a missed test will require 2 months for a subsequent result. Although this timing is still better than in-clinic reviews every six months, we feel it is too long for detection of nAMD and treatment.

Our participants were given specific instructions to undertake MRFh routinely on a suitable day/time of the week in a quiet room; however, the distractions of daily life cannot always be avoided and may negatively impact test results. Despite this, we find high test reliability of 96% with MRFh compared to 98% in-clinic with MRFc. This aligns with our findings in a cohort of high-risk glaucoma participants who undertook MRFh over a 12-month period [10] and those of other research groups who utilized ‘free-space’ viewing with the MRF [23]. Note that fixational losses could not be monitored by the MRF application as the iPad mini screen dimensions are too small to present stimuli to the blind spot.

One limitation of this study is the small *n* of the study group. As a result, we were unable to investigate any differences due to demographics, e.g., was compliance better in older vs. younger participants? Another limitation is the order of the MRF tests that were administered by the application. Testing HVA first at 135 cd/m^2^ may have the effect of bleaching the retina, thus providing sub-optimal results for the subsequent LLVA and RS measures. A better method might be to start with RS and follow it with LLVA and HVA.

In this study, we investigated MRFh monitoring of RS and two types of VA (HVA and LLVA) with in-clinic measures. We found MRFh outcomes in RS were 2 dB better than when conducted on the MAIA. This likely reflects differences in test background (5 vs. 1.3 cd/m^2^) as is evident from the data provided by Vingrys et al. [24], which we previously reported in our short-term MRFh publication [9].

An interesting observation from the participant perceptions survey is that, even though only 50% completed the study, most iAMD patients found MRFh easy to use (Q1, Figure 6). A subgroup analysis of those who completed the study (Likert score: 4.6 [4.9, 4.3]) versus those that did not (Likert score: 4.6 [4.8, 4.4]) found no difference (*p* > 0.05). Therefore, the authors believe that personal factors, such as competing life demands and the MRF being too much effort to fit into their daily schedule, were reasons for not completing the study rather than the test being too difficult to use. An independent survey of a West African population reported that 92% of patients with glaucoma and 88% of control patients indicated that the MRF was simple to use [25].

HM with MRFh returned a false positive rate of 7% (*n* = 3/46), triggering an alert to the study coordinator due to the confirmed change in RS. This compares to a false alert rate of 21% in the AREDS2-Home study with the PHP device [13], although another study reports false positive alerts to be as high as 93% at one clinical site with the same device [2]. Whilst there were no cases of progression from iAMD to nAMD in this study, the relatively lower rate of alerts justifies the need for larger clinical trials with the MRFh device.

## 5. Conclusions

This 12month MRFh study in participants with iAMD finds compliance to weekly testing of 61% with a hightest reliability of 96%. Study retention dropped to 50% by the end of the study with lack of motivation being a major driver to drop out. MRFh vision testing application shows promise in monitoring patients with iAMD between scheduled clinical visits; however, addressing long term compliance is an issue that needs further consideration. Additional studies are needed to determine efficacy in detecting progression to nAMD in large prospective clinical trials.

## Figures and Tables

**Figure 1 jcm-12-02530-f001:**
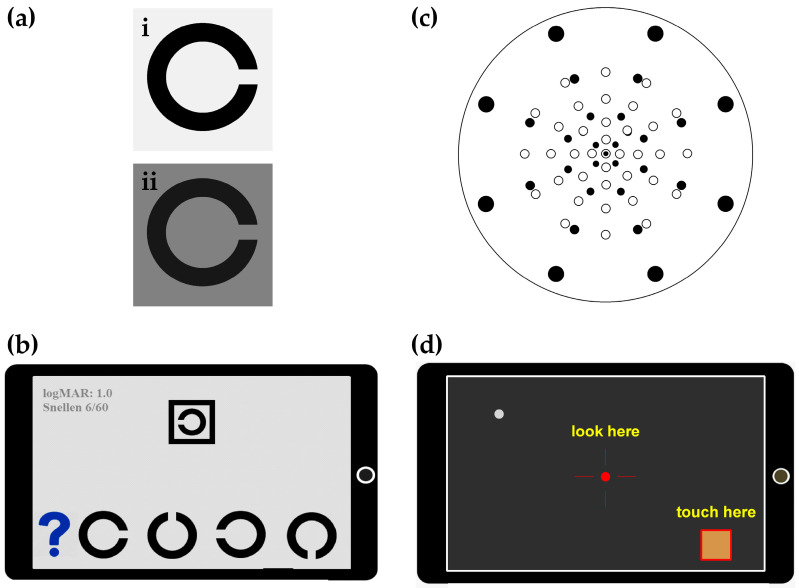
The Melbourne Rapid Fields vision testing application. (**a**) The visual acuity Landolt C targets. (**i**) High contrast visual acuity (HVA, L_b_: 135 cd/m^2^, contrast: 94%). (**ii**) Low-luminance visual acuity (LLVA, L_b_: 5 cd/m^2^, contrast: 86%). (**b**) Patients adopt a 4-alternate-forced-choice to identify the orientation of the target shown in the central box. (**c**) Visual field protocol. Unfilled circles represent MAIA test spot locations and filled circles denote the MRF grid. Note size scaling of MRF stimuli occurs away from fixation. Black ring is 10° eccentricity. (**d**) The visual field task. App-generated audio prompts instruct participants to look at the red fixation cross and tap in the designated ‘touch zone’ when a stimulus is observed.

**Figure 2 jcm-12-02530-f002:**
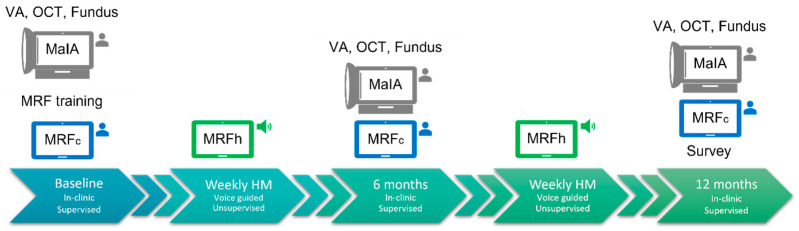
Timeline for the 12-month voice-guided telemedicine study outlined in the text. VA: Early Treatment Diabetic Retinopathy Study acuity chart viewed under normal clinical conditions (HVA) and through an ND2 filter (LLVA). OCT: Optical coherence tomography. Fundus: Dilated fundus exam. MAIA: Macular Integrity Assessment. MRFc: Melbourne Rapid Fields visual acuity and visual field testing performed in-clinic under supervision (cartoon of person at these times). MRFh: Melbourne Rapid fields visual acuity and visual fields performed at-home with app-generated audio prompts (cartoon of speaker at these times). Survey: Survey of participant perceptions of self-monitoring after 12-months.

**Figure 3 jcm-12-02530-f003:**
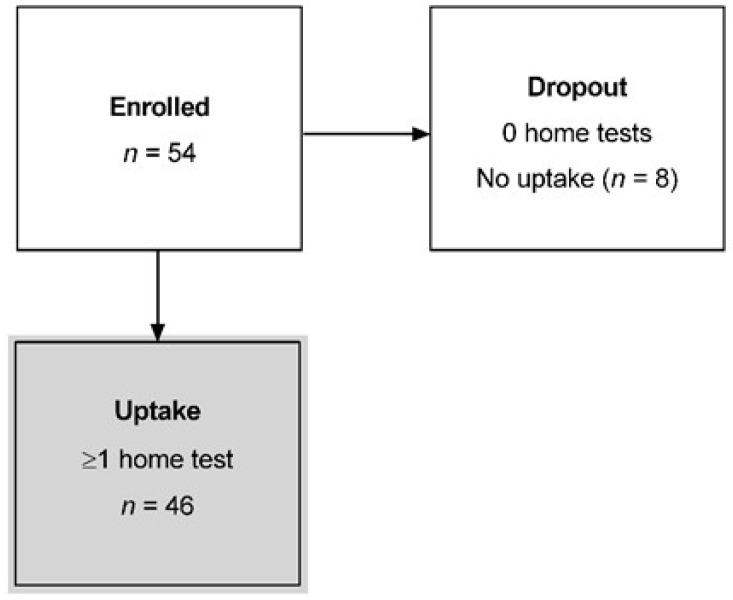
CONSORT diagram for the long-term Melbourne Rapid Fields home (MRFh) study. *n* = 54. Shaded box indicates the number of participants with uptake.

**Figure 4 jcm-12-02530-f004:**
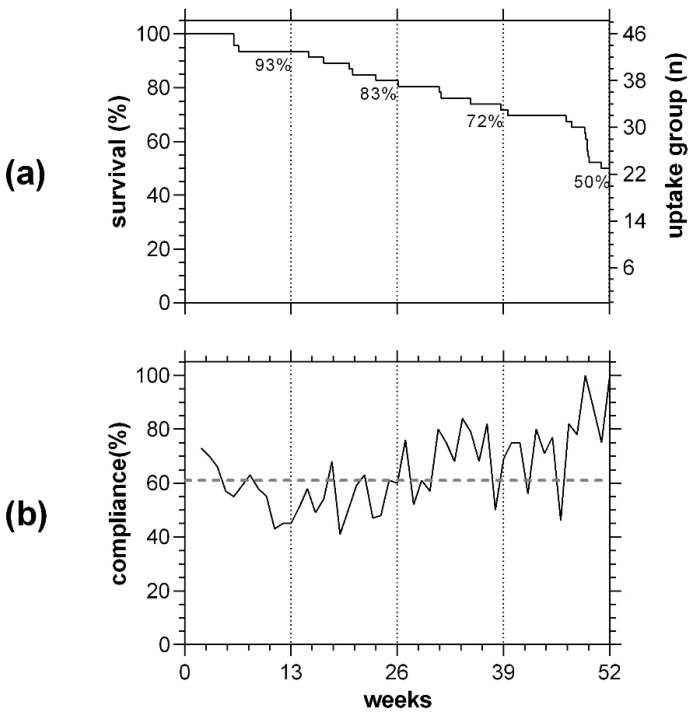
Survival and compliance of *n* = 46 iAMD participants to the request for MRFh over a 52-week period. (**a**). Survival of participants over 52 weeks of MRFh. (**b**). Compliance rate to weekly testing (7 + 1 days, solid line). Dotted line represents average compliance rate over the 52-week period.

**Figure 5 jcm-12-02530-f005:**
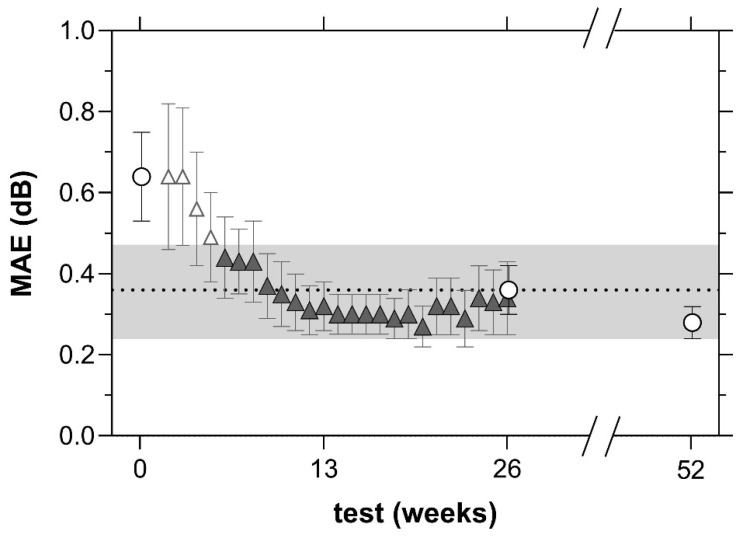
Mean absolute error (MAE, dB) from the median retinal sensitivity calculated from the average of 3 MAIA tests performed in-clinic (circles) and 26 MRFh tests performed at home (triangles) in *n* = 46 participants with intermediate AMD. The unfilled triangles are shown from week 2 and represent a period of learning where the MAE approaches its plateau. Error bars show SEM. The black dotted line shows the MAE of the MAIA at week 26. Grey zone represents 2x SEM of the 6-month MAIA result.

**Figure 6 jcm-12-02530-f006:**
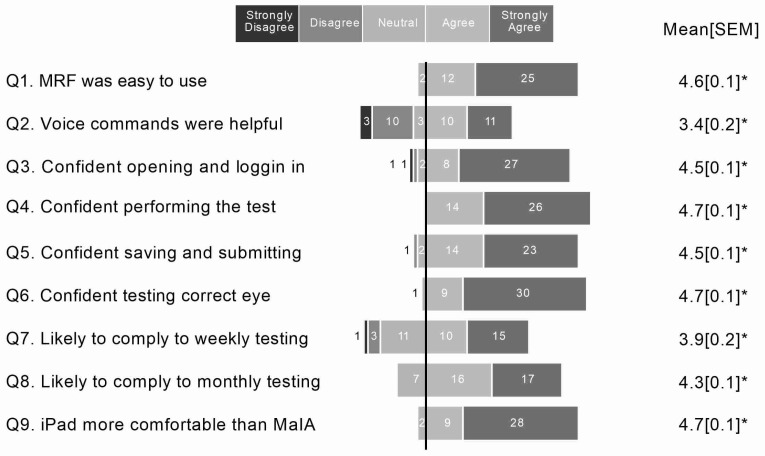
Survey responses on the ease of use of the MRF iPad application from *n* = 40 participants with iAMD. * Indicates mean is significantly removed from the neutral state of 2.5, *p* < 0.01. MAIA = Macular Integrity Assessment.

**Table 1 jcm-12-02530-t001:** Patient demographics.

	Total Group, *n* (Range)
Test subjects	54
Age, y (min–max)	67 (51–82)
Gender (female)	82%

**Table 2 jcm-12-02530-t002:** Test return frequency from *n* = 46 intermediate age-related macular degeneration participants with uptake of home monitoring.

Days	Count (*n*)	Freq (%)	Freq (% cumul.)
0–8	813	61	61
9–15	286	21	82
16–22	130	10	92
23–29	54	4	96
>29	55	4	100
Total	1338		

Number of home exams (*n*) returned for each of the specified periods. Shaded area represents result submitted after follow-up reminder (email).

**Table 3 jcm-12-02530-t003:** Percentage of reliable tests over 52-weeks.

	Reliable, % (SD)
MAIA	100 (0)
MRFc	98.9 (7.5)
MRFh	95.9 (7.4) ***

Criteria for a reliable test for MRF are false positives and false negatives ≤25%. Criterial for a reliable test for MAIA is false positives ≤25%. *** Indicates significantly different from MAIA, *p* < 0.001.

**Table 4 jcm-12-02530-t004:** Group averages (SD in brackets) for visual acuity (ETDRS vs. MRFh) and visual field (MAIA vs. MRFh) after 12-months of home monitoring.

Visual Field (dB)	MAIA	MRFc	MRFh
RS	27.1 (1.2)	29.4 (0.7) ***	29.1 (1.8) ***
Visual acuity (letters)	ETDRS	MRFc	MRFh
HVA	86.6 (4.3)	86.3 (7.5)	87.9 (3.1)
LLVA	72.4 (5.7)	81.4 (9.1) ***	81.5 (6.9) ***

ETDRS: Early treatment diabetic retinopathy study. MAIA: Macular Integrity Assessment microperimeter. MRFc: Melbourne Rapid Fields conducted in clinic with clinical assistant: Melbourne Rapid Fields conducted at home under voice-guidance. All MRF testing was at near with reading glasses as needed. RS: Retinal sensitivity. HVA: High contrast visual acuity. LLVA: Low luminance visual acuity. Significance of *** indicates that MRF (in clinic or at home) is significantly different to the in-clinic assessment (MAIA or ETDRS) at the *p* < 0.001 level. There was no significant difference between the supervised (MRFc) or unsupervised (MRFh) MRF outcomes.

## Data Availability

A publicly archived dataset generated during the study is available at the following DOI: https://doi.org/10.6084/m9.figshare.22130702.v1.

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
