# Peer review of "Performance of a Smart Device over 12-Months for Home Monitoring of Patients with Intermediate Age-Related Macular Degeneration"

_jcm, 2023, doi:10.3390/jcm12072530_

Round 1
Reviewer 1 Report
The authors would like to develop a cost-effective approach to monitor patients aiming to detect the progression from iAMD to nAMD (I would prefer the term exudative AMD as neovascular AMD does not have to be exudative)
In general, the manuscript is well written but my main concern is how the data was presented. Specific comments as follows:
1. Compliance and retention: The authors presented data on overall compliance rate of about 50%, but there were only 26 out of the the initial 46 participants completed the 12 months study. So is the 50% all from these 26 patients. It does not appear to be. Similarly, there were data to show when the test results were reported but I am not sure how that is useful. I wondered whether the authors would like to consider what would be the minimial and ideal frequency of tests, and how many patients out of the 46 managed the minimal and / or ideal frequency. For instance, for this test to be useful, we should expect the participants to do the test at least once every 4 weeks, how many would have managed that. That would be more useful data. I apprecipate we don't know the ideal frequency at the moment, but at least, we would know what to aim for and what is practical in a larger study.
2. False positive alerts: The authors mentioned that there were 7% of false positive alerts. Is that 3 out of 46 participants? However, only 26 got to the end of study. So why 3/46? Bearing in mind, a false positive leads to anxiety and urgent clinic visit. I understand that this is a pilot study but pilot study is to generate data to see whether a bigger study is justified, so this rate should be discussed.
3. The survey were received on 40 participants. I assume it was only requested at the end of the study, but only 26 completed the study at 12 months. I am a bit sceptial of the results when say Q1 MRF was easy to use which most people strongly agree. If so, why only 26 completed the study, or that 25 strongly agree were all those who completed?
4. I think PHP is a daily test, so in the discussion, when the authors were comparing compliance, it might not be fair to compare daily vs weekly test.
5. Minor points:
a) Might be worthwhile to know roughly how long does it take to do that test.
b) It is well known that there could be OCT changes without visual acuity changes. Is there evidence of correlation between MRFh alerts to onset of exudative AMD without symptoms? As the study is of the better seeing eye, so if there is visual acuity changes, one would assume the patient would have symptoms, so to be useful, it needs to show MRFh alerts occur when there were no symptoms. That could have be published elsewhere but did not see that mention in the manuscript.
Author Response
Please see attached word document.

Reviewer 2 Report
The clinical work, “Performance of a Smart Device over 12-months for Home Monitoring of Patients with Intermediate Age-Related Macular Degeneration” targeting to practical application assess the ability of MRFh to detect early progression to nAMD.
The comments are:
1. Please mention the number of patients involve in study, in abstract and where ever possible.
2. Why tested to access to an iPad? Is there any particular reason?
3. As previous comments, Each clinical visit included ETDRS VA (HVA and LLVA with ND2 filter) at 6m? Please verify.
4. Line 172: Study completion, a survey of perceptions of home monitoring, comprising 9 questions. Is this questionably based on experience and literature?
Is there any possibility of biases in understating when training, how to use it?
Author Response
Please see attached word document.

Round 2
Reviewer 1 Report
I am happy with this version although I think the study design could be improved by including high risk patients. As no one converted over 12 months, these patients might not be the target patients to be used. Nonetheless, that cannot be changed now.